# Electron recombination of rotationally cold $D_2H^+$ ions

A. Znotins [1], A. Faure [2], C. H. Greene [3], M. Grieser [1], F. Grussie[1], L. W. Isberner [1,4], Á. Kálosi [1,5], V. Kokoouline[6], D. Müll[1], D. Paul [1,5], M. Pezzella[7], D. W. Savin [5], S. Schippers [4], J. Tennyson [7], A. Wolf [1], O. Novotný [1] & H. Kreckel [1] ✉

Dissociative recombination (DR) of electrons with small molecular ions is a fundamental process for the physics and chemistry of the interstellar medium and planetary atmospheres. In previous DR studies, detailed analysis of the experimental rate coefficients has been hindered by the difficulty of preparing the ions in well-defined quantum states. For polyatomic ions in particular, truly state-selective measurements have been elusive, allowing only qualitative benchmarks of theory. Here, we present DR studies of the deuterated triatomic hydrogen ion $D_2H^+$, where the molecular ions were stored for up to 1000 seconds inside the Cryogenic Storage Ring (CSR) prior to the DR measurements. Our experiments with rotationally cold $D_2H^+$ ions allow for detailed comparison to state-of-the-art theoretical calculations. We obtain very good agreement between experiment and theory even in the important collision energy range from 1 meV to 0.5 eV, where a multitude of Rydberg resonances reveal their imprint on the rate coefficient.

When gas is partially ionized by radiation or energetic particles, the ionization balance is determined by neutralization processes that are dominated by the recombination of positively charged atomic or molecular ions with free electrons. For atomic ions, the only efficient mechanism to release the recombination energy and stabilize the neutral (in the absence of a third collision partner) is the emission of photons, rendering atomic recombination inefficient for plasmas at moderate density and temperature. When molecular ions capture an electron, on the other hand, the excess energy can lead to the break-up of the excited complex into neutral fragments, in a process called dissociative recombination (DR)[1]. The DR rate of molecular ions is often orders of magnitude faster than the radiative recombination rate of atomic ions. Consequently, the recombination of molecular ions plays a crucial role in many environments, ranging from the interstellar medium[2–4] to Earth's ionosphere[5], the atmosphere of Venus, and by

extension, of exoplanets[6], and technical plasmas relevant for the design of fusion reactors[7].

The theoretical treatment of the DR process is a challenging many-body problem that is complicated by the fact that an infinite number of Rydberg states of the neutral molecule have to be taken into account. The most successful method to manage the complexity is the multichannel quantum defect theory (MQDT)[8,9], which combines dissociative states into so-called *channels*, thereby limiting the necessary couplings to a finite number. While this approach has yielded very good results for a number of diatomic ions[10–12], the full-dimensional treatment of the DR of polyatomic ions, complicated by the need to account for the various possible initial states of excitation, remains a formidable challenge. In this framework, the DR of the triatomic hydrogen ion $H_3^+$ —the simplest polyatomic molecule – has long served as a benchmark for theoretical studies, which have identified the

[1]Max-Planck-Institut für Kernphysik, Heidelberg, Germany. [2]IPAG, Université Grenoble Alpes, CNRS, Grenoble, France. [3]Department of Physics and Astronomy, Purdue University, West Lafayette, IN, USA. [4]I. Physikalisches Institut, Justus-Liebig-Universität Gießen, Gießen, Germany. [5]Columbia Astrophysics Laboratory, Columbia University, New York, NY, USA. [6]Department of Physics, University of Central Florida, Orlando, FL, USA. [7]Department of Physics and Astronomy, University College London, London, UK. ✉e-mail: holger.kreckel@mpi-hd.mpg.de

Jahn–Teller effect as the main driver of the DR process for $H_3^+$ at low temperature[13–15].

The experimental determination of DR rate coefficients comes with its own challenges. While more than thirty DR measurements with triatomic hydrogen ions can be found in the literature, the results have been rather controversial at times, as various techniques have yielded rate coefficients that differed by orders of magnitude[16,17]. One of the main issues is the control of the internal state of the ions prior to the recombination measurements. Since standard ion sources usually produce molecular ions in highly excited vibrational and rotational states, it is crucial to find a way to confine the internal population to well-defined quantum states for meaningful experiments. For this reason, the heavy ion storage ring technique has become the gold standard for DR measurements[18]. The main advantages over traditional plasma techniques lie in the essentially background-free environment that the high vacuum of the storage ring affords, combined with long measurement times that allow for radiative cooling of molecular ions with a permanent dipole moment. While initial studies in room temperature storage rings enabled measurements with vibrationally cold ions, the importance to control the rotational excitation has since been recognized. However, reducing the rotational degrees of freedom requires much lower temperatures, enabled only by the recent development of cryogenic storage rings, which allow for experiments with superior internal state control, down to individual quantum states. The most methodological of such studies have been carried out at the cryogenic storage ring (CSR) in Heidelberg, Germany (Fig. 1a). Stored inside a cryogenic storage ring, diatomic molecular ions with a permanent dipole moment can cool to their lowest rotational states[19–21], enabling quantum state-selective DR measurements of important species such as $HeH^+$ [22], $CH^+$ [23] and $OH^+$ [24] by direct detection of the recombination products (Fig. 1b).

While the above studies with diatomic ions mark a new era for experimental DR studies, the problem with measuring the DR of cold $H_3^+$ arises from the existence of excited metastable rotational states, owing to the triangular symmetry of the molecule[25,26]. It still poses a considerable difficulty to prepare an ensemble of $H_3^+$ ions with well-defined internal excitation[27–29]. However, $D_2H^+$ ions possess a permanent dipole moment of 0.48 D. This enables cooling by spontaneous emission of radiation; and, as we will show below, the population distribution of initially hot $D_2H^+$ ions can be reduced to a handful of identifiable rotational states within a few hundred seconds of storage inside the CSR (Fig. 1c). Therefore, DR measurements with $D_2H^+$ in a cryogenic storage ring allow for experiments with superior internal-state definition, facilitating an independent cross-check of the theoretical approach that is used for $H_3^+$, where the exact internal excitation is much harder to assess[27,28].

Apart from its fundamental relevance, the DR of triatomic hydrogen has attracted a lot of attention because of its relevance for the chemistry of interstellar clouds[3,30,31], where $H_3^+$ ions are the drivers of an active ion–neutral chemistry network[32]. The deuterated isotopologues $H_2D^+$ and $D_2H^+$ have been detected in the denser regions of interstellar clouds, where stars and planets are being formed[33–35]. In these environments, the DR of deuterated triatomic hydrogen ions releases D atoms, which in turn engage in surface reactions with more complex species that are frozen out on dust grains[34]. This leads to an enhancement of the observed deuterium fraction in large molecules (e.g., up to 13 orders of magnitude for $CH_3OH$[36]), which can be used to trace organic chemistry in these dense cores and examine possible links with solar system deuterium reservoirs. Thus, detailed knowledge of the low-temperature DR rate coefficients for deuterated triatomic hydrogen ions is crucial for our understanding of the deuterium fractionation in star-forming regions.

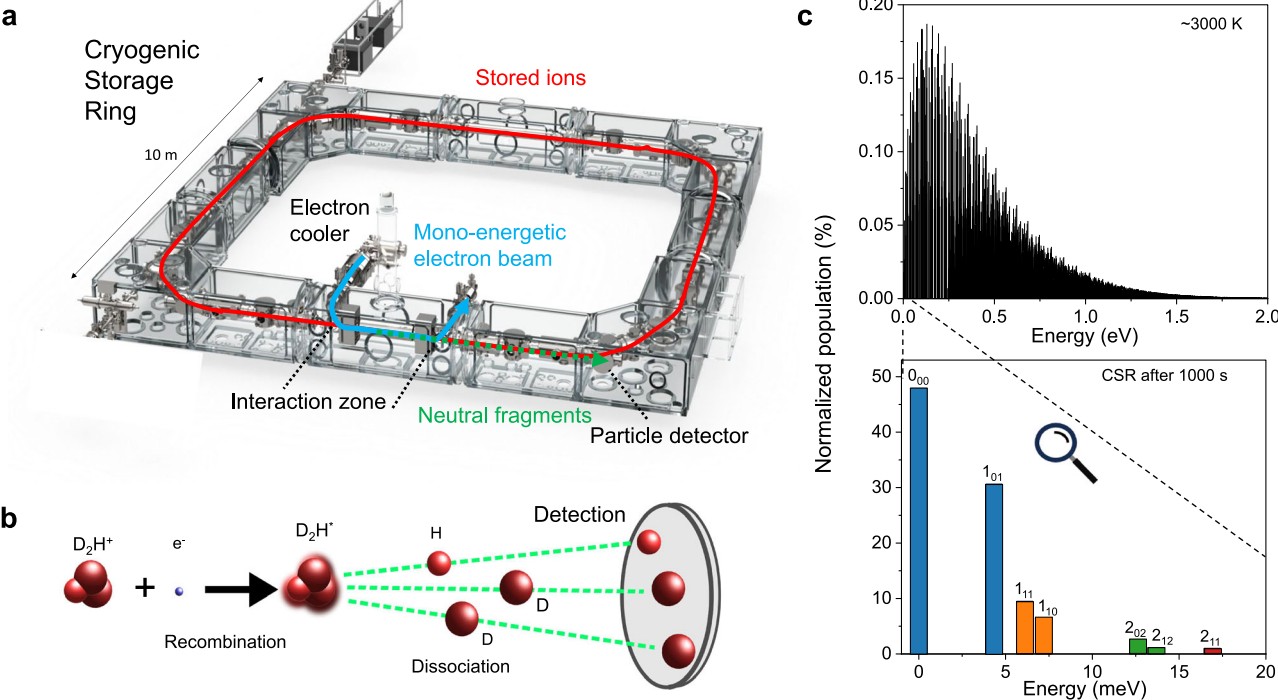

**Fig. 1 | Overview of the cryogenic storage ring (CSR), the DR process, and the effect of internal cooling of $D_2H^+$. a** Schematic of the CSR with the electron cooler, interaction zone, and neutral particle detector locations noted. The CSR features a nested vacuum structure, with a large outer cryostat and inner ultra-high vacuum chambers that are cooled by liquid helium to temperatures below 6 K. **b** Schematic of the recombination event, showing the molecular break-up and detection of the neutral fragments. **c** Upper panel: Stick diagram of the estimated initial population distribution of the rovibrational states of $D_2H^+$. Assuming a temperature of 3000 K, the population is dispersed over hundreds of individual states. Lower panel: Simulated populations for $D_2H^+$ after storage of 1000 s inside the cryogenic vacuum of the CSR, with a combined ~80% of the population ending up in the two ground states of the respective nuclear spin manifolds ($0_{00}$ for ortho-$D_2H^+$, and $1_{01}$ for para-$D_2H^+$).

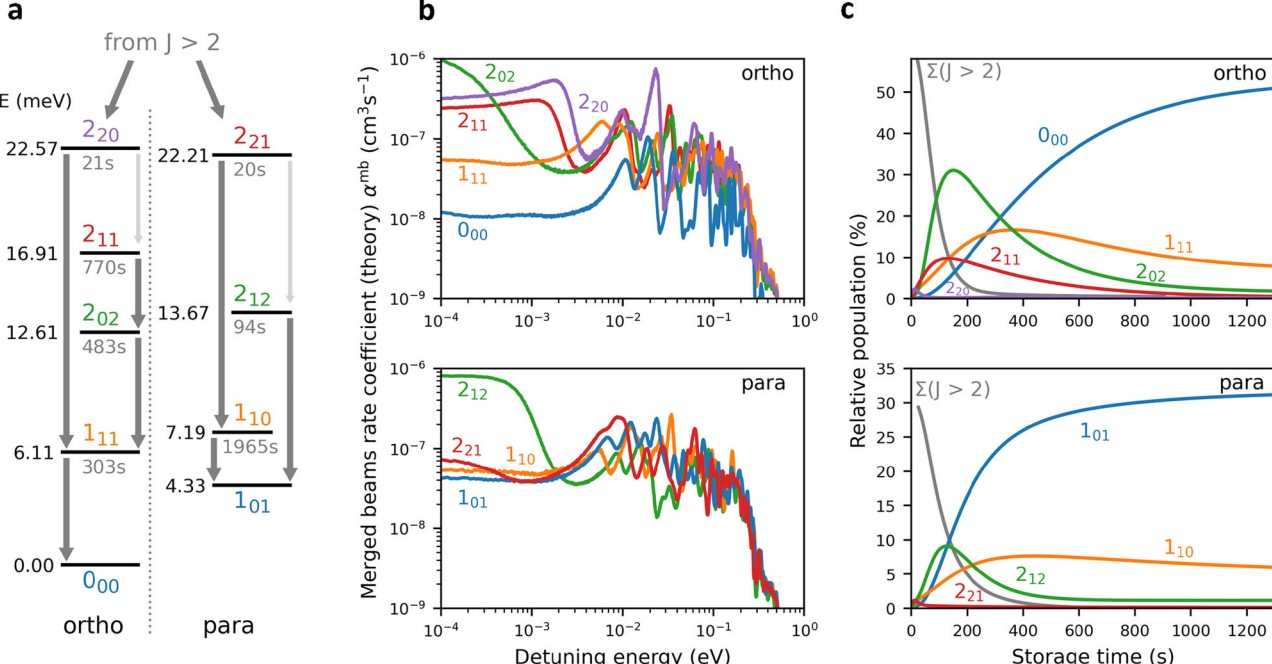

**Fig. 2 | Energy level scheme, calculated state-specific DR rate coefficients, and predicted cooling curves. a** Energy level scheme and radiative transitions for all $D_2H^+$ states with $J < 3$. The states are sorted into the respective ortho and para manifolds, which are strictly separated. The labels in color denote the rotational quantum numbers $J_{K_aK_c}$. The lifetimes of the excited states are given in gray below each level. The individual energies are given to the left of each level (in meV). The dark gray arrows indicate major radiative transitions, while the thinner arrows in light gray indicate minor transitions (where applicable). **b** Individual theoretical merged-beams DR rate coefficients $\alpha^{mb}$ for the lowest rotational states. The curves were derived from the theoretical DR cross sections (see Supplementary Information), which were convolved with the experimental velocity distribution for comparison to the experimental results. **c** Simulated relative populations for the rotational states of $D_2H^+$ as a function of storage time within the CSR. The initial temperature of the ensemble was set to 3000 K. While the initial distributions are spread over hundreds of rovibrational states, here we plot only the lowest rotational states with $J < 3$, which remain populated after 1000 s of storage. Ortho and para states are plotted separately. The population of all states with $J > 2$ is summed up and depicted by the gray solid lines.

## Results

### Internal cooling of stored $D_2H^+$ ions

Ensembles of molecular ions extracted from a standard ion source feature a high degree of internal excitation, and it is common to assume temperatures in the range of several thousand kelvin[24,42]. Consequently, the initial population is dispersed among hundreds of excited states. To assess the degree of rovibrational cooling for $D_2H^+$ as

Previous studies of the DR of $D_2H^+$ yielded ambiguous results. Room temperature measurements at the TSR facility found a rate coefficient that exceeded the theoretical rates by an order of magnitude at the lowest energies[37,38]. The experimental results were corroborated by measurements at the CRYRING facility and the theoretical DR calculations were called into question[39]. Afterglow studies carried out between 80 and 145 K[40,41], on the other hand, agreed better with the theoretical calculations for $D_2H^+$, but those measurements also found no discernible difference between the various isotopologues of triatomic hydrogen, which was found to be in conflict with the theoretical predictions. However, all of the above experimental studies were carried out either at elevated temperature (storage rings) or density (afterglow); conclusions on the validity of the theoretical approach should therefore be treated cautiously.

Here, we present both experimental and theoretical electron recombination rate coefficients for $D_2H^+$ in defined quantum states. The experiments were carried out at the CSR, employing the velocity-matched beam of the CSR low-energy electron cooler for recombination experiments in a collinear arrangement (Fig. 1a). We combine the DR studies with a comprehensive model for the evolution of the internal state population of $D_2H^+$ during storage inside the cryogenic vacuum of the CSR.

a function of storage time inside the CSR, we prepared a master equation model (see "Methods" section for details) that follows the evolution of all internal states of the molecule during storage. The state energies and radiative transition strengths are based on the most recent ExoMol calculations[43,44], taking into account 50,000 states connected by 40 million transitions. As the dominant cooling process is the spontaneous emission of radiation, it is instructive to consider the radiative lifetimes of the lowest rotational states (Fig. 2a).

$D_2H^+$ exists in two nuclear spin configurations, denoted *ortho* and *para*, which cannot be inter-converted by radiative transitions or inelastic electron collisions (see Methods section for nuclear spin configurations and rotational quantum numbers of $D_2H^+$). Therefore, even for asymptotic radiative cooling to the ambient 6 K field, the population will be shared among at least two states, namely the ortho and para ground states $J_{K_aK_c} = 0_{00}$ and $1_{01}$, respectively. The most long-lived excited state (para-$1_{10}$) has a lifetime of 1965 s, and some of the ortho states have lifetimes of several hundred seconds. The lifetimes of higher-lying states with $J > 2$ tend to decrease with increasing energy, as more decay channels become available.

Additional cooling can be achieved by inelastic collisions with low-energy electrons inside the CSR electron cooler[45]. We have calculated cross sections for rotational de-excitation (as well as excitation) of $D_2H^+$ by electron collisions (see Supplementary Information for details). We include the corresponding rates in our model for transitions among states with $J \le 3$. Similar to radiative transitions, inelastic electron collisions do not change the nuclear spin of the molecules.

Finally, the relative populations can be influenced by the depletion of individual states through the DR process itself. Figure 2b shows our DR calculations (details can be found in the Supplementary Information) for all states with $J < 3$. The calculated DR cross sections

were convolved with the experimental energy resolution to yield the equivalent of *merged-beams* rate coefficients $\alpha^{mb}$ for the present experiment, as a function of the relative electron–ion collision energy. The theoretical DR rate coefficients are incorporated as a loss process in our master equation model. A more complete description of the model and the new calculations for inelastic and dissociative electron collisions is given in the Methods section and the Supplementary Information.

Figure 2c shows the calculated evolution of all rotational states with $J < 3$ for characteristic measurement conditions inside the CSR. As can be seen, we expect most of the population (~80%) to be accumulated in the respective nuclear spin ground states (blue lines) after 1000 s of storage. For both configurations an additional 5–10% of the population is found in the first excited states (yellow lines). The gray lines depict the sum of the populations in higher lying states ($J > 2$), which dominate at early times, but become insignificant after storage times >500 s. The dominant cooling process remains radiative emission, while inelastic electron collisions also result in a net cooling effect, increasing the final populations in the ground states by up to 10%. The selective depletion of individual states by DR has a smaller impact on the relative populations, but it is the only process that affects the ortho:para ratio, which slowly shifts away from the canonical 12:6 value once DR depletion is included in the calculations.

Figure 2c represents the prediction of the evolution of the internal states of $D_2H^+$ under realistic measurement conditions inside the CSR, based on state-of-the-art calculations for all relevant processes. Our updated DR calculations (Fig. 2b) predict marked differences between individual states, and a trend toward lower rate coefficients with decreasing $J$ quantum number is seen, most notably for the ortho configurations. As the population in the low-$J$ states builds up during storage, we expect the total measured DR rate coefficient to decrease for long storage times. Furthermore, the individual DR rate coefficients show pronounced structures between 1 meV and 0.5 eV. In all likelihood these structures will be washed out at early storage times, as the population is smeared out over hundreds of states, but they should emerge at long storage times, when the population becomes concentrated in the lowest rotational states.

**Dissociative recombination measurements**

The CSR is a fully electrostatic storage ring with a circumference of 35 m (Fig. 1a). It employs a nested vacuum structure, where the inner experimental chambers can be cooled to liquid helium temperature (≤6 K). Surface condensation combined with dedicated entrapment pumps leads to very low residual gas number densities (~$10^3$ cm$^{-3}$), facilitating storage times of up to hours for heavier ion beams (see ref. 46 for details).

The $D_2H^+$ ions were produced in a duoplasmatron ion source and electrostatically accelerated to a kinetic energy of 250 keV. Mass selection was achieved by a dipole magnet in the CSR transfer line. Unlike the case of $H_2D^+$, where contaminations with $D_2^+$ ions are known to be problematic, the production of pure $D_2H^+$ beams is straightforward, due to the absence of molecular isobars with a weight of 5 u in a pure hydrogen/deuterium discharge. We stored ~1–3 × $10^8$ $D_2H^+$ ions per injection.

A dedicated electron cooler in one of the straight sections of the ring is used to inject a nearly mono-energetic electron beam into the CSR, where it is merged with the stored ions over an effective electron–ion overlap length of 0.72 m. At matched electron–ion velocities, the electron beam can be used for phase-space cooling of the stored ion beam[47], while the electron velocity can also be detuned to perform energy-resolved DR studies, covering several orders of magnitude in relative collision energy. The neutral DR fragments are detected by a micro-channel plate detector, situated downstream of the electron target.

As the DR process can limit the lifetime of the ion beam in the ring, great care was taken to adapt the electron density and measurement scheme such that beam depletion by DR was kept at an acceptable level. We devised a storage cycle with a measurement phase at very early storage times (4–9 s) and another measurement phase at long storage times (1000–1300 s) to study the effect of internal cooling. The electron beam intensity was reduced at intermediate times to avoid excessive depletion by DR, while maintaining phase space cooling of the ion beam at all times. Between 1 and 5 × $10^6$ $D_2H^+$ ions remained in the ring after 1000 s of storage.

Providing absolute values for the experimental DR rate coefficients requires the determination of the number of ions in the ring as well as measurements of the electron density, which are carried out routinely at the CSR[23,24,45]. We assume a total systematic uncertainty of 20% for the absolute scale of the measurements, with the main contributions arising from the ion current determination (12.5%) and the uncertainty in electron density (10%).

Figure 3a shows the experimentally measured merged-beams DR rate coefficients

$$\alpha^{mb}(E_d) = \langle \sigma v_r \rangle_{E_d}, \tag{1}$$

for $D_2H^+$ at storage time intervals of 4–9 s (red markers) and 1000–1300 s (black markers). The merged-beams rate coefficient is the product of the energy-dependent DR cross section $\sigma$ and the relative electron–ion velocity $v_r$, averaged over the experimental velocity distribution in the electron–ion interaction region. The latter is also influenced by systematic effects in the beam merging and de-merging regions, which have to be accounted for in the convolution of the theoretical cross sections when comparing experimental and theoretical results[23]. The rate coefficient is plotted as a function of the detuning energy $E_d$, defined as the nominal center-of-mass collision energy for an electron beam at energy $E_e$ with respect to the laboratory frame energy of the electron beam $E_0$ at matched ion-electron velocities

$$E_d = \left( \sqrt{E_e} - \sqrt{E_0} \right)^2. \tag{2}$$

The comparison between short and long storage times confirms the predicted effect of internal cooling, manifested in an overall decrease of the low-energy rate coefficient by a factor of 3–6. For the data at 1000–1300 s, we can infer the relative populations of individual rotational states from the master equation model (Fig. 2c). Consistent with the dominance of only very few states, pronounced resonances become visible between 1 meV and 0.5 eV. Within the MQDT calculations, these features result from the indirect DR process, where the electron is captured resonantly into a neutral Rydberg state of the molecule followed by coupling to a dissociative state. For these particular storage conditions, we can directly weight the theoretical rate coefficients (Fig. 2b) with the final state populations that result from the master equation approach for the late storage times, to obtain the blue line in Fig. 3a.

A comparison with the theoretical results must consider the difficulty of predicting electronic excitation energies, as required in the theoretical calculations, with an accuracy of <0.01 eV in the ab-initio framework of the MQDT approach. We find excellent agreement on the absolute scale at collision energies of >20 meV, while the theoretical calculations underestimate the rate coefficient at the lowest energies. This deviation, however, is caused by a few individual Rydberg resonances at very low electron collision energies, whose predictions suffer most from the uncertainty of the ab-initio potential surfaces available to the MQDT model. The logarithmic energy scale over-emphasizes these low energies. It is important to note that at $E_d$ < 2 meV the velocity spread of the electron beam becomes

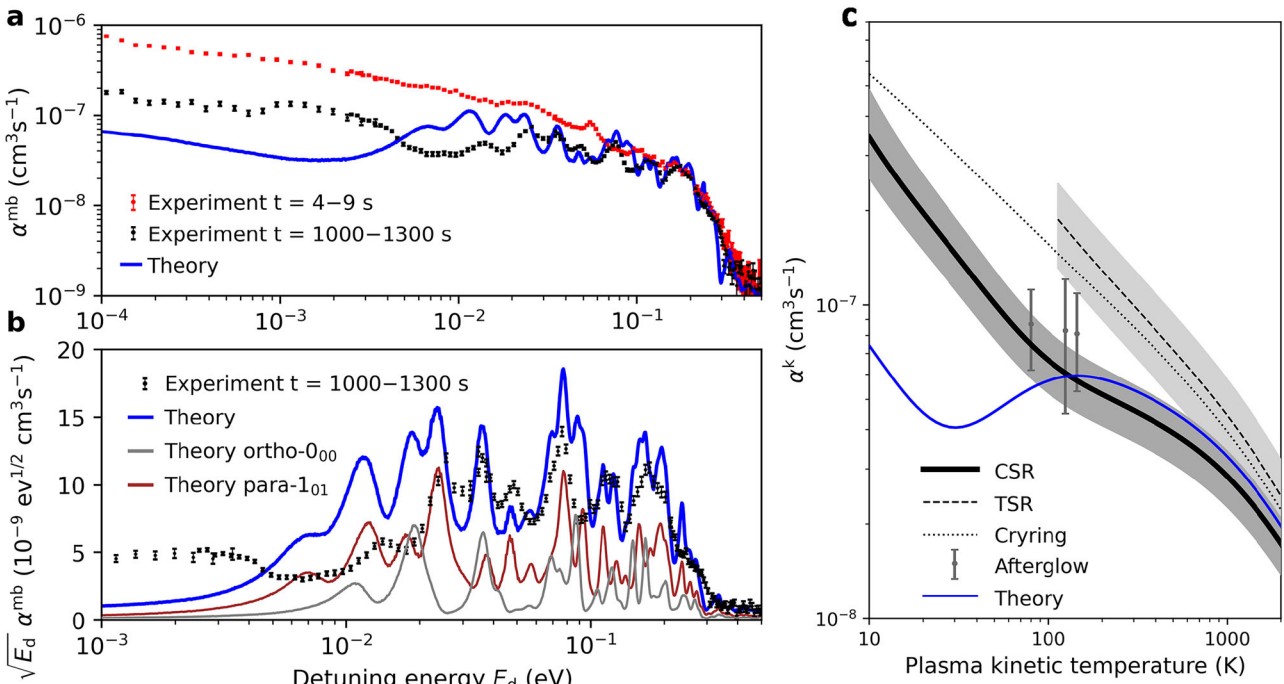

**Fig. 3 | Dissociative recombination rate coefficients for $D_2H^+$. a** Measured merged-beams DR rate coefficient $\alpha^{mb}$ at storage times 4–9 s (red markers) and 1000–1300 s (black markers). The blue solid line shows the theoretical rate coefficients weighted with the simulated state populations for 1000–1300 s of storage. The experimental error bars represent statistical 1-sigma uncertainties only, while the systematic uncertainty of the absolute scale is 20%. **b** Reduced merged-beams DR rate coefficients $\sqrt{E_d}\,\alpha^{mb}$ for storage times of 1000–1300 s (black markers). The blue solid line depicts the theoretical rate coefficients weighted with the simulated state populations for 1000–1300 of storage. The gray and brown lines

show the individual population-weighted contributions of the lowest ortho- and para-states, respectively. **c** Plasma kinetic temperature rate coefficient $\alpha^k$ derived from the cold measurements (CSR), compared to previous measurements (TSR[37], Cryring[39], Afterglow[41]). The shaded areas around the CSR and TSR results represent the total (systematic and statistical) uncertainties of these measurements, which have been propagated through the conversion procedure Supplementary. For comparison, the blue line depicts the plasma kinetic temperature rate coefficient derived from the population-weighted theoretical calculations.

significant, such that the merged-beams rate coefficient $\alpha^{mb}$ is influenced by the instrumental line shape. However, this effect does not account for the discrepancies between experiment and theory, as the velocity spread is taken into account for the calculation of the theoretical curves.

## Discussion

To facilitate a more detailed examination of the resonant structures, Fig. 3b shows the reduced DR rate coefficient[28], given as the product of the merged-beams rate coefficient and the square root of the detuning energy $\sqrt{E_d}\,\alpha^{mb}$. This representation serves to eliminate the intrinsic $1/\sqrt{E}$ threshold dependence of the rate coefficients at low energy[48,49], and it allows for a meaningful representation of the rate coefficient on a linear scale. In addition to the state-weighted theory, also the predicted contributions of the most-populated, lowest ortho-$0_{00}$ and para-$1_{01}$ states are shown (cf. Fig. 2c). Striking similarity of the calculated and measured structures is found at >20 meV, even suggesting that the para-$1_{01}$ state dominates the DR rate for certain of the measured peaks. The remaining discrepancies at lower collision energies likely indicate the intrinsic uncertainties of the ab-initio electronic excitation energies entering the calculations. All together, the present comparison of state-of-the-art experiment and theory—enabled by the superior internal state-definition that cryogenic storage affords – indicates very promising pathways for future state-selective studies with polyatomic molecular ions.

Since the DR reactions studied here are important for astrochemical models and technical plasma, we have converted our experimental data into a kinetic temperature rate coefficient. The procedure and analytical fits are described in the Supplementary Information. Figure 3c shows the rate coefficient $\alpha^k$ as a function of the

gas kinetic temperature $T_k$ compared to previous experimental results. Our results indicate a much slower recombination rate for $D_2H^+$ at low temperature compared to the two room temperature storage ring studies[37,39], which both differ from the present measurements by a factor of ~3 at 100 K. Within the error margins our data agree with the previous afterglow studies[41]. However, in contrast to afterglow experiments conducted at comparatively high pressure, we can definitely rule out ternary effects—which can have a large impact on the recombination rate.

The kinetic temperature rate coefficient derived from the state-weighted theoretical data is in very good agreement with the CSR result down to a kinetic temperature of 100 K. The comparison in Fig. 3c shows that the plasma rate coefficients derived from room-temperature storage-ring measurements for kinetic temperatures below 1000 K were significantly affected by contributions from excited rotational states.

The lower DR rate observed in the present study for cold $D_2H^+$ suggests an enhanced survival rate with respect to DR in interstellar environments. This may have great relevance for the deuterium fractionation in star forming regions. The present study therefore highlights the risk of extrapolation from room temperature data, and the need to measure important astrophysical reactions under accurately simulated interstellar conditions.

## Methods

### Rotational quantum numbers and nuclear spin symmetry for $D_2H^+$

The $D_2H^+$ ion is an asymmetric top molecule, belonging to the $C_{2v}$ point group. The total rotational angular momentum quantum number (excluding spin) is denoted $J$. By convention, the three principal axes of

rotation $(a, b, c)$ are chosen such that the rotational constants are in the order $B_a > B_b > B_c$. The projections of the angular momentum onto the two axes with the extreme moments of inertia are denoted $K_a$ and $K_c$, respectively. We use rotational state labels $J_{K_a K_c}$. As an example: $1_{01}$ stands for $J = 1$, $K_a = 0$ and $K_c = 1$.

The nuclear spin $I$ of the molecule imposes restrictions on the rotational quantum numbers through the symmetry requirements of the total wave function, which has to be symmetric with respect to the permutation of the two identical deuterons. Consequently, the nuclear spin designations follow closely the case of the $D_2$ molecule. The nuclear spin symmetries and the corresponding restrictions for the rotational quantum numbers are summed up in Table 1.

## Master equation approach to model the time evolution of the rovibrational states of $D_2H^+$

The production of triatomic hydrogen ions in partially ionized plasma proceeds through the classical ion-neutral reaction

$$H_2 + H_2^+ \longrightarrow H_3^+ + H, \tag{3}$$

which is exothermic by 1.732 eV[50]. The exothermicity changes only slightly for reactions of deuterated variants, e.g., the reaction of $H_2^+$ with HD forming $H_2D^+$ has an exothermicity of 1.726 eV[50]. Recent studies show that a significant fraction (about 2/3) of this excess energy will end up as internal excitation of the triatomic hydrogen molecules after the reaction[51]. Some of the excitation may be quenched in subsequent collisions in the ion source, but even for carefully optimized conditions, the internal temperature of triatomic hydrogen ions extracted from standard discharge ion sources will always exceed 1000 K[52].

In our experiments, the molecular ions are stored in the cryogenic vacuum of the CSR for up to 1000 s prior to the electron recombination measurements. During the storage time the ions cool by spontaneous emission of radiation. Furthermore, interactions between the ions and electrons inside the electron cooler may change the internal state of the ions, either by inelastic collisions or by depletion of individual states through the DR process. In order to make the best possible prediction for the time evolution of the internal states, we employ state-of-the-art theoretical calculations for all relevant processes.

We use a master equation approach to simulate the time evolution of the internal excitation of an ensemble of initially hot $D_2H^+$ ions during the DR measurements inside the CSR. This requires solving the following set of coupled ordinary differential equations

$$\begin{aligned}
\frac{dP_i(t)}{dt} = &- P_i(t) \sum_j A_{ij} + \sum_k P_k(t) A_{ki} \\
&+ \sum_j P_j(t) B_{ji} \rho(\nu_{ji}) - \sum_k P_i(t) B_{ik} \rho(\nu_{ik}) \\
&+ \sum_j P_j(t) \alpha_{ji}^{\text{in}} n_e - \sum_k P_i(t) \alpha_{ik}^{\text{in}} n_e \\
&- \sum_j P_i(t) \alpha_{ij}^{\text{in}} n_e + \sum_k P_k(t) \alpha_{ki}^{\text{in}} n_e \\
&- P_i(t) \alpha_i^{\text{DR}} n_e,
\end{aligned} \tag{4}$$

where $P_i$ denotes the population of an individual molecular state $i$ as a function of storage time $t$. The radiative processes are represented by, respectively, the Einstein $A$ and $B$ coefficients for spontaneous decay and absorption (as in previous studies of internal cooling, we neglect the stimulated emission process[25]). The indices $k$ and $j$ denote molecular states situated energetically higher and lower than state $i$, respectively, while the index $i$ covers all molecular states included in the model. Furthermore, $\nu$ denotes the frequency of a particular transition, and $\rho(\nu)$ the spectral density of the radiation field. The rate coefficients for state-changing inelastic collisions with electrons are

**Table 1 | Nuclear spin configurations and symmetry restrictions for the rotational projection quantum numbers $K_a$ and $K_c$ of the $D_2H^+$ ion**

| Configuration | Ortho (12) | Para (6) |
|---|---|---|
| D symmetry | $A_1$ | $B_2$ |
| Nuclear spin | $I_D = 0(1)$ | $I_D = 1(3)$ |
| | $I_D = 2(5)$ | |
| H symmetry | $A_1$ | $A_1$ |
| Nuclear spin | $I_H = 1/2(2)$ | $I_H = 1/2(2)$ |
| Rot. quantum numbers | $K_a + K_c$ even | $K_a + K_c$ odd |

The nuclear spin $I_D$ of the deuteron pair and the nuclear spin of the proton $I_H$ are considered separately, since hyperfine interactions are expected to be negligible here. The numbers in parentheses give the multiplicities.

denoted $\alpha^{\text{in}}$, while the theoretical prediction for state-specific DR rate coefficients are given by $\alpha^{\text{DR}}$. The electron density is denoted by $n_e$.

In line with previous studies[22,45], we assume a two-component blackbody radiation field inside the CSR, where 99% of the effective surface area is represented by the cold inner chamber walls at 6 K, while the second component with a fraction of 1% accounts for room temperature radiation leaks from viewports and openings into the inner chambers. This effective radiation field defines the spectral energy density $\rho(\nu)$ for a given transition frequency. The Einstein coefficients are taken from the most recent ExoMol line list calculations[43,44]. The model encompasses a line list for $D_2H^+$ containing 50,000 rovibrational states, interconnected by a network of 40 million transitions. The model is sufficient to accommodate the initial condition of a 3000 K Boltzmann distribution, which serves as an estimate of the average internal energy of ions generated by the duoplasmatron ion source (the exact value of the initial temperature has very limited impact on the final state populations, as long as it exceeds 1000 K).

State-changing inelastic electron collisions are included through the relevant $J \to J'$ transitions among the lowest rotational states. The rate coefficients of electron impact excitation $\alpha_{J \to J'}(E)$ as a function of collision energy $E$ were obtained from dedicated cross-section calculations, performed by combining R-matrix calculations with the fixed-nuclei (FN) and Coulomb–Born (CB) approximations (see Supplementary Information), which were folded with our experimental electron velocity distribution. The corresponding de-excitation cross sections $\sigma_{J' \to J}(E)$ were calculated using detailed balance.

We considered only state-changing collisions among states with $J \le 3$. Since the electron density in our experiment was comparatively low ($1 \times 10^5 \text{ cm}^{-3}$), and the overlap inside the electron target is only a small fraction of the circumference of the storage ring, the effective rate for inelastic electron collisions is on the order of $10^{-4} \text{ s}^{-1}$ or less. As $D_2H^+$ ions (unlike $H_3^+$) do not possess metastable states, the radiative transitions among all higher-lying states are faster by orders of magnitude compared to inelastic electron collisions. Consequently, radiative processes dominate the internal cooling at early times, however, the effect of inelastic electron collisions does influence the final state populations by up to 10% for $D_2H^+$.

The model also incorporates population losses due to state-specific DR of the lowest rotational states, employing cross sections from our updated DR calculations (see Supplementary Information). Depletion by DR was included for all states with $J < 3$; its main effect is a slow depopulation of the states in the ortho manifold at intermediate times, mainly due to the comparatively high DR rate coefficients and long lifetimes predicted by theory for the $J = 2$ ortho rotational states.

The outcome of the master equation including all processes is depicted in Figs. 1c and 2c of the main manuscript. The model predicts that only six states carry significant populations after 1000 s of storage. The ortho-states carry more population than the para-states, which is consistent with their larger statistical weights (the canonical

ortho-to-para ratio based on statistical degeneracies is 12:6 for $D_2H^+$, see Table 1).

## Data availability

The molecular line list data for $D_2H^+$ are available on the ExoMol website (www.exomol.com). Kinetic temperature rate coefficients for inelastic electron collisions will be made available through the EMAA (Excitation of Molecules and Atoms for Astrophysics) database (https://doi.org/10.17178/EMAA). Analytic fit functions for the kinetic temperature DR rate coefficients for $D_2H^+$ are given in the Supplementary Information. Experimental and theoretical merged-beams DR rate coefficients are available from the authors upon request.

## Code availability

The R-matrix code (UKRMol+) used for inelastic electron collision calculations is publicly available at (https://doi.org/10.17632/k3ny7zcfrb.1). The codes used for the line list calculations are available on the ExoMol website (www.exomol.com).

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

## Acknowledgements

This work was supported by the Max Planck Society. We thank Z. Mašín for providing details on his ASYMTOP code. CHG is supported by the U.S. Department of Energy, Office of Science, Basic Energy Sciences, under Award No. DE-SC0010545. The work performed at UCL was supported by the European Research Council (ERC) under the European Union's Horizon 2020 research and innovation programme through Advanced Grant number 883830. A.K., D.P. and D.W.S. were supported, in part, by the National Science Foundation Astronomy and Astrophysics Grants Program under AST-1907188 and the NASA Astrophysics Research and Analysis Program under 80NSSC19K0696 and 80NSSC24K0206. L.W.I and S.S. are grateful for financial support by the Deutsche Forschungsgemeinschaft via project 431145392.

## Author contributions

A.Z., O.N., A.W. and H.K. designed and planned the experiment. A.Z., M.G., F.G., L.W.I., Á.K., D.M., D.P., O.N. and H.K. performed the storage ring experiments. A.Z., Á.K., D.P. and O.N. analyzed the experimental DR data. A.F. and J.T. calculated the cross sections for inelastic electron collisions. V.K. and C.G. performed the theoretical DR cross-section calculations. A.Z., O.N. and H.K. simulated the internal state evolution of $D_2H^+$ with the aid of M.P. D.W.S. and S.S. assisted with the data analysis. The manuscript was written by A.Z., A.F., V.K. and H.K. All authors reviewed the manuscript and commented on the manuscript.

## Funding

## Competing interests

The authors declare no competing interests.
