## [Transparent Peer Review file · Nature Communications]

Electron recombination of rotationally cold D_2H^+ ions

Corresponding Author: Dr Holger Kreckel

Version 0:

Reviewer comments:

Reviewer #1

(Remarks to the Author)

Referee report for the manuscript entitled:

Electron recombination of rotationally cold D_2H^+ ions

by A. Znotins, A. Faure, C.H. Greene, M. Grieser, F. Grussie, L.W. Isberner, A. Kálosi, V. Kokouline, D. Müll, D. Paul, M. Pezzella, D.W. Savin, S. Schippers, J. Tennyson, A. Wolf, O. Novotný, and H. Kreckel

Abstract:

Dissociative recombination (DR) of electrons with small molecular ions is a fundamental process for the physics and chemistry of both the interstellar medium and planetary atmospheres. In previous DR studies, detailed comparisons between experimental rate coefficients and theory have been hindered by the difficulty of preparing the ions in well-defined quantum states prior to the recombination measurements. For polyatomic ions in particular, where the theoretical treatment of the DR process becomes considerably more complex, truly state-selective measurements have been elusive, allowing only qualitative benchmarks of theory. Here, we present DR measurements of the deuterated triatomic hydrogen ion D_2H^+ , performed at the Cryogenic Storage Ring (CSR). The molecular ions were stored for up to 1000 s inside the cryogenic vacuum of the CSR prior to the measurements, allowing them to cool to their lowest rotational states. Our DR measurements with D_2H^+ are unprecedented in internal-state definition compared to previous experiments with any polyatomic molecular ion, permitting detailed comparison with state-of-the-art theoretical calculations. We obtain very good agreement between experiment and theory even in the important collision energy range from 1 meV to 0.5 eV, where a multitude of Rydberg resonances reveal their imprint on the rate coefficient.

In the manuscript the dissociative recombination measurements of the deuterated triatomic hydrogen ion, D_2H^+ was reported, that was performed on the Cryogenic Storage Ring at Heidelberg Germany.

The molecular ions were stored for up to 1000 s at cryogenic temperatures in ultrahigh vacuum in the CSR prior to the measurements, allowing them to cool to their lowest rotational (ortho and para) states of the target. This was supported by predictions made on the evolution of the internal states of D_2H^+ under realistic measurement conditions inside the CSR, based on the solution of the master equation approach using state-of-the-art calculations for all relevant processes. The measured convoluted cross sections are compared with the best adapted multichannel quantum defect calculations performed with a slightly modified version, that has been successfully applied for the lightest triatomic hydrogen molecular cation.

Excellent agreement was found on absolute scale between experiment and theory even in the important collision energy range from 20 meV to 0.5 eV, where a multitude of Rydberg resonances are governing the cross section.

The convoluted cross sections are then converted in an iterative way to thermal rate coefficients. The obtained results indicate a much slower recombination rate for D_2H^+ at low temperature compared to the previous room temperature storage ring and afterglow studies. The kinetic temperature rate coefficient derived from the state-weighted theoretical data is in very good agreement with the CSR results down to about 100 K kinetic temperature.

The final conclusion is, that the lower DR rate observed for cold D_2H^+ suggests an enhanced survival rate with respect to DR in interstellar environments.

I believe that the presented experimental results obtained with CSR and the excellent agreement obtained with the state-of-

the-art MQDT calculations makes the manuscript ready for publication in Nature Communications.

Here follows a short list of queries to the authors:

- 1.) Do the authors plan to perform similar measurements for the other triatomic isotopologues of hydrogen to describe in details the isotopic effect for the simplest triatomic molecular cation.
- 2.) Could the authors estimate/quantify in what extent are responsible the electron structure and electron scattering (quantum chemistry) calculations responsible for the less good agreement between the CSR and MQDT convoluted cross sections below 20 meV? Is the displacement (between experiment and theory) of the isolated resonances at low collision energy responsible for the larger difference between the CSR and MQDT thermal rate coefficients? What possible ways of improvement (theoretical or numerical) can be imagined for the low-energy parts?
- 3.) Have the authors applied the present improved MQDT method for the previously studied H₃⁺ benchmark system? If yes, how it compares the two sets of cross sections?
- 4.) On page 6 middle-part of the first column:
"For these particular storage conditions, we can directly weigh the theoretical rate coefficients (Fig. 2b) with the final state populations that result from the master equation approach for the late storage times, to obtain the blue line in Fig. 3a."
I would change weigh to weight.

Reviewer #2

(Remarks to the Author)

Report on the article 'Electron recombination of rotationally cold D₂H⁺ ions' by A.Znotins et al.

This article describes a huge step forward to the detailed understanding and modelling of the mechanisms driving the dissociative recombination (DR) and the electronic impact ro-vibrational transitions (RVT) of H₃⁺ and its isotopologues, in this case on D₂H⁺.

It also proves the huge potential of the CSR storage ring techniques on one hand and of the MQDT, of the R-matrix and of the theoretical spectroscopy methods on the other hand in revealing these mechanisms with high accuracy.

The particularly long lifetime of the beam - a success in itself - insures to achieve the dominance of a few states of D₂H⁺ with $J < 3$ after 1000 s of the injection. This is remarkable, since the initial population distribution corresponding to 3000 K spreads over hundreds of rovibrational states.

Given the exceptionally low temperature of 6K in the cryogenic vacuum chamber, the experiment results in involving almost exclusively (more than 80%, at late enough time), the ortho and para ground states, allowing an unprecedentedly reached state-to-state monitoring of the DR for triatomic systems.

The time-evolution of the relative populations for the rotational states of D₂H⁺ as a function of storage time within the CSR is simulated. This is a real challenge when one targets ultra-high accuracy. It became possible due to the huge theoretical recent progress: very accurate up-to-date data on energy levels and radiative lifetimes computed and recorded in the EXOMOL data base, as well as increasingly accurate cross sections and rate coefficients for DR and RVT coming from advanced MQDT and R-matrix calculations, all of these injected in a very detailed and comprehensive master equation system.

As for these latter DR and RVT theoretically computed rate coefficients (coming from Maxwell isotropic and anisotropic averaging), they agree very well with the experimental ones on the whole range of energy/temperature. Moreover - and this is a further notable advancement - this stands in particular for the VERY low energy range 1-500 meV which, in the past, was object of disagreement in different approaches, being dominated by numerous Rydberg resonances, difficult to measure and to model.

Taken into account the numerous decisive steps forward made by the CSR team in the measurement techniques and by the R-matrix and MQDT experts in the theoretical modeling, I strongly recommend the publication of this paper in Nature Communication.

Reviewer #3

(Remarks to the Author)

The paper describes an unprecedented state resolved DR study of the deuterated trihydrogen ion, a central system to the modeling of the ISM astrochemistry. It is also a benchmark system for exploring processes in polyatomic molecules. The work successfully compares state of the art experimental measurements with new theoretical calculations (both for the state-resolved DR itself and for the cooling of the stored D₂H⁺), thus correcting earlier estimates of this ions destruction rate. I therefore support the publication of this work in nature communications.

Nevertheless, I have few comments and queries that I think will be valuable also for the interested readers:

1. In the discussion of the theory-exp differences at low E_d (pg 6), the authors imply that the differences could be due to the finite spread in the velocity distribution. However, the spread already convoluted with the theoretical prediction? I would expect this to result in a smearing and a plateau, but not the observed step in the rate at $\sim 4\text{meV}$.
2. Considering the state resolved theoretical DR rates (with $J=2$ states exhibit a step at $\sim 4\text{meV}$), is it possible to fit the contribution of $J=2$ states and explain the experimental step as an excess of $J=2$ states, beyond what is predicted based on the master eq? Thus providing an explanation that does not require assuming a "missing" low energy resonance in the theoretical treatment?
3. The conclusion that the 100K thermal rate coefficient is substantially lower, compared with previous storage ring studies is very interesting. The authors mention that it may affect the D/H ratio. However, it would be important for the readers to know if this means that also the previous storage ring based thermal rate coefficients for undeuterated H_3^+ could be similarly overestimated? What would be the potential impact on ISM modeling?

Minor comments:

4. The authors emphasize that ortho-para could change due to DR and by not radiative decay. Would be nice also to explicitly comment about the effect of inelastic collisions in the main text. Also, could inelastic collisions at the early stages of storage with high J states cause ortho-para transitions and potentially affect the asymptotic state distribution at longer times?
5. On pg. 2, authors provide references for earlier spectroscopic studies that demonstrate cooling of ions in cryogenic storage devices. It might be good to consider citing also the measurements in other devices (e.g. DESIREE) to give the broader context of other storage devices.
6. At the bottom of pg 5, the authors state that D_2H^+ has no molecular isobars. I guess that the authors expect HeH^+ not to be present in the source ? I can be a reasonable assumption, but I suggest to rephrase the statement.

Version 1:

Reviewer comments:

Reviewer #1

(Remarks to the Author)

Dear Editor,

The authors have answered in detail all my questions so I warmly suggest the publication of the manuscript in Nature Communications in the present form.

Reviewer #2

(Remarks to the Author)

I am satisfied with the changes.

Reviewer #3

(Remarks to the Author)

The authors addressed all my questions and concerns.

I recommend publishing the manuscript as is.

Response letter:**Manuscript „ Electron recombination of rotationally cold D2H+ ions”
Nature Communications**

On behalf of the authors, we wish to thank all three referees for their thorough review of our manuscript and the positive comments. Please find our replies to the individual queries below (we quote the reviewers' comments in blue color).

Reviewer #1 (Remarks to the Author):

Referee report for the manuscript entitled:

Electron recombination of rotationally cold D2H+ ions

by A. Znotins, A. Faure, C.H. Greene, M. Grieser, F. Grussie, L.W. Isberner, A. Kálosi, V. Kokoouline, D. Müll, D. Paul, M. Pezzella, D.W. Savin, S. Schippers, J. Tennyson, A. Wolf, O. Novotný, and H. Kreckel

Abstract:

Dissociative recombination (DR) of electrons with small molecular ions is a fundamental process for the physics and chemistry of both the interstellar medium and planetary atmospheres. In previous DR studies, detailed comparisons between experimental rate coefficients and theory have been hindered by the difficulty of preparing the ions in well-defined quantum states prior to the recombination measurements. For polyatomic ions in particular, where the theoretical treatment of the DR process becomes considerably more complex, truly state-selective measurements have been elusive, allowing only qualitative benchmarks of theory. Here, we present DR measurements of the deuterated triatomic hydrogen ion D2H+, performed at the Cryogenic Storage Ring (CSR). The molecular ions were stored for up to 1000 s inside the cryogenic vacuum of the CSR prior to the measurements, allowing them to cool to their lowest rotational states. Our DR measurements with D2H+ are unprecedented in internal-state definition compared to previous experiments with any polyatomic molecular ion, permitting detailed comparison with state-of-the-art theoretical calculations. We obtain very good agreement between experiment and theory even in the important collision energy range from 1 meV to 0.5 eV, where a multitude of Rydberg resonances reveal their imprint on the rate coefficient.

In the manuscript the dissociative recombination measurements of the deuterated triatomic hydrogen ion, D2H+ was reported, that was performed on the Cryogenic Storage Ring at Heidelberg Germany.

The molecular ions were stored for up to 1000 s at cryogenic temperatures in ultrahigh vacuum in the CSR prior to the measurements, allowing them to cool to their lowest rotational (ortho and para) states of the target. This was supported by predictions made on the evolution of the internal states of D2H+ under realistic measurement conditions inside the CSR, based on the solution of the master equation approach using state-of-the-art calculations for all relevant processes.

The measured convoluted cross sections are compared with the best adapted multichannel quantum defect calculations performed with a slightly modified version, that has been successfully applied for the lightest triatomic hydrogen molecular cation.

Excellent agreement was found on absolute scale between experiment and theory even in the important collision energy range from 20 meV to 0.5 eV, where a multitude of Rydberg resonances are governing the cross section.

The convoluted cross sections are then converted in an iterative way to thermal rate coefficients. The obtained results indicate a much slower recombination rate for D₂H⁺ at low temperature compared to the previous room temperature storage ring and afterglow studies. The kinetic temperature rate coefficient derived from the state-weighted theoretical data is in very good agreement with the CSR results down to about 100 K kinetic temperature.

The final conclusion is, that the lower DR rate observed for cold D₂H⁺ suggests an enhanced survival rate with respect to DR in interstellar environments.

I believe that the presented experimental results obtained with CSR and the excellent agreement obtained with the state-of-the-art MQDT calculations makes the manuscript ready for publication in Nature Communications.

Here follows a short list of queries to the authors:

1.) Do the authors plan to perform similar measurements for the other triatomic isotopologues of hydrogen to describe in details the isotopic effect for the simplest triatomic molecular cation.

We very much appreciate the interest in comprehensive studies for the other isotopologues. Indeed, studies with H₂D⁺ are underway. The preparation of a pure H₂D⁺ ion beam is experimentally more challenging, as contaminations with D₂⁺ ions have to be quantified and eventually avoided.

Furthermore, experiments toward laser-spectroscopy of H₃⁺ inside the CSR are being prepared, and we hope that this approach will enable state-selective studies with H₃⁺ in the future.

2.) Could the authors estimate/quantify in what extent are responsible the electron structure and electron scattering (quantum chemistry) calculations responsible for the less good agreement between the CSR and MQDT convoluted cross sections below 20 meV? Is the displacement (between experiment and theory) of the isolated resonances at low collision energy responsible for the larger difference between the CSR and MQDT thermal rate coefficients? What possible ways of improvement (theoretical or numerical) can be imagined for the low-energy parts?

Unfortunately, we cannot give a definitive answer as to what causes the low-energy discrepancy between experiment and theory. The present calculations use an energy-independent scattering matrix, and it is conceivable that this limits the accuracy of the theoretical approach at low energies. As the experimental resolution is also limited at the lowest collision energies, it is difficult to compare individual resonance positions at a few meV collisions energy. For the future, it is foreseen to adapt the energy-dependent frame transformation approach [in the spirit of Hvizdos, Curik, Greene, PRA 111, 012805 (2025)] to the various triatomic hydrogen isotopologues, and we hope that this will shed light on this issue.

3.) Have the authors applied the present improved MQDT method for the previously studied H₃⁺ benchmark system? If yes, how it compares the two sets of cross sections?

We have tested the present level of theory for the H₃⁺ system for a few rotational states, and the comparison to previous experimental studies (at TSR) did not change much. However, one has to bear in mind that the internal state-definition of the previous H₃⁺ measurements was poor compared to the present experiments with D₂H⁺. We hope that future state-selective experiments

with H3+ (using laser-probing and manipulation of the individual state populations) at CSR will allow for a much more detailed experiment-theory comparison.

4.) On page 6 middle-part of the first column:

“For these particular storage conditions, we can directly weigh the theoretical rate coefficients (Fig. 2b) with the final state populations that result from the master equation approach for the late storage times, to obtain the blue line in Fig. 3a.”

I would change weigh to weight.

Done.

Reviewer #2 (Remarks to the Author):

Report on the article 'Electron recombination of rotationally cold D2H+ ions' by A.Znotins et al.

This article describes a huge step forward to the detailed understanding and modelling of the mechanisms driving the dissociative recombination (DR) and the electronic impact ro-vibrational transitions (RVT) of H3+ and its isotopologues, in this case on D2H+.

It also proves the huge potential of the CSR storage ring techniques on one hand and of the MQDT, of the R-matrix and of the theoretical spectroscopy methods on the other hand in revealing these mechanisms with high accuracy.

The particularly long lifetime of the beam - a success in itself - insures to achieve the dominance of a few states of D2H+ with $J < 3$ after 1000 s of the injection. This is remarkable, since the initial population distribution corresponding to 3000 K spreads over hundreds of rovibrational states.

Given the exceptionally low temperature of 6K in the cryogenic vacuum chamber, the experiment results in involving almost exclusively (more than 80%, at late enough time), the ortho and para ground states, allowing an unprecedentedly reached state-to-state monitoring of the DR for triatomic systems.

The time-evolution of the relative populations for the rotational states of D2H+ as a function of storage time within the CSR is simulated. This is a real challenge when one targets ultra-high accuracy. It became possible due to the huge theoretical recent progress: very accurate up-to-date data on energy levels and radiative lifetimes computed and recorded in the EXOMOL data base, as well as increasingly accurate cross sections and rate coefficients for DR and RVT coming from advanced MQDT and R-matrix calculations, all of these injected in a very detailed and comprehensive master equation system.

As for these latter DR and RVT theoretically computed rate coefficients (coming from Maxwell isotropic and anisotropic averaging), they agree very well with the experimental ones on the whole range of energy/temperature. Moreover - and this is a further notable advancement - this stands in particular for the VERY low energy range 1-500 meV which, in the past, was object of disagreement in different approaches, being dominated by numerous Rydberg resonances, difficult to measure and to model.

Taken into account the numerous decisive steps forward made by the CSR team in the measurement techniques and by the R-matrix and MQDT experts in the theoretical modeling, I strongly recommend the publication of this paper in Nature Communication.

No changes to the manuscript required.

Reviewer #3 (Remarks to the Author):

The paper describes an unprecedented state resolved DR study of the deuterated trihydrogen ion, a central system to the modeling of the ISM astrochemistry. It is also a benchmark system for exploring processes in polyatomic molecules. The work successfully compares state of the art experimental measurements with new theoretical calculations (both for the state-resolved DR itself and for the cooling of the stored D_2H^+), thus correcting earlier estimates of this ions destruction rate. I therefore support the publication of this work in nature communications.

Nevertheless, I have few comments and queries that I think will be valuable also for the interested readers:

1. In the discussion of the theory-exp differences at low E_d (pg 6), the authors imply that the differences could be due to the finite spread in the velocity distribution. However, the spread already convoluted with the theoretical prediction? I would expect this to result in a smearing and a plateau, but not the observed step in the rate at $\sim 4\text{meV}$.

The referee is correct. Indeed, the main effect of the electron velocity distribution is a smearing of features at very low energies. The goal of our comment was to point out that a fraction of Figure 3a is affected by the velocity spread and that this region may appear overemphasized because of the logarithmic representation. We feel that we do need to mention this effect for fair representation of the experimental data. However, we have now modified this paragraph and we clarify that the velocity spread is taken into account for the theoretical curves already, and that it can NOT explain the discrepancies between experiment and theory.

2. Considering the state resolved theoretical DR rates (with $J=2$ states exhibit a step at $\sim 4\text{meV}$), is it possible to fit the contribution of $J=2$ states and explain the experimental step as an excess of $J=2$ states, beyond what is predicted based on the master eq? Thus providing an explanation that does not require assuming a “missing” low energy resonance in the theoretical treatment?

That is a very interesting question. We have tried a “free” fit of the experimental rate coefficient with all available theoretical rate coefficients. However, the fit did not improve the comparison. In fact, the low energy shape of the experimental rate coefficient does not exactly match the predicted curves for $J=2$, and their inclusion invariably leads to discrepancies at higher energies. Therefore, we decided that the most appropriate comparison within the scope of the present publication is between the experimental rate and the best available theoretical prediction based on populations derived from state-of-the-art calculations of all relevant processes.

3. The conclusion that the 100K thermal rate coefficient is substantially lower, compared with previous storage ring studies is very interesting. The authors mention that it may affect the D/H ratio.

However, it would be important for the readers to know if this means that also the previous storage ring based thermal rate coefficients for undeuterated H₃⁺ could be similarly overestimated? What would be the potential impact on ISM modeling?

We believe that it is difficult to draw conclusions from our measurements with D₂H⁺ to the experimental results for H₃⁺. Previous storage ring measurements with D₂H⁺ clearly suffered from the influence of room temperature radiation inside the old generation of storage rings. This is different for H₃⁺ ions, which couple only very inefficiently to the blackbody radiation field. Therefore, the present results do not invalidate previous H₃⁺ measurements performed using rotationally cold ion sources. Nevertheless, we do aim for state-selective DR measurements with H₃⁺ ions in the future, applying direct laser-probing of individual rotational states inside the storage ring.

Minor comments:

4. The authors emphasize that ortho-para could change due to DR and by not radiative decay. Would be nice also to explicitly comment about the effect of inelastic collisions in the main text. Also, could inelastic collisions at the early stages of storage with high J states cause ortho-para transitions and potentially affect the asymptotic state distribution at longer times?

We have added a sentence in the main text, explaining that inelastic collisions leave the nuclear spin unchanged.

Concerning inelastic collisions of high-J states: all highly excited states decay fast, when compared to our experimental time scale. Since we operated the electron cooler with comparatively low electron density, we expect that inelastic collisions do not play a role for those states (as described in the Methods section). And as nuclear spin-changing electron collisions would have to rely on higher order effects, ortho-para transitions should be suppressed even more.

5. On pg. 2, authors provide references for earlier spectroscopic studies that demonstrate cooling of ions in cryogenic storage devices. It might be good to consider citing also the measurements in other devices (e.g. DESIREE) to give the broader context of other storage devices.

We have made the statement more general and included a reference to [Schmidt, PRL 119, 073001 (2017)].

6. At the bottom of pg 5, the authors state that D₂H⁺ has no molecular isobars. I guess that the authors expect HeH⁺ not to be present in the source ? I can be a reasonable assumption, but I suggest to rephrase the statement.

Very good point. We re-phrased, now it reads: "... due to the absence of molecular isobars with a weight of 5u in a pure hydrogen/deuterium discharge".